# TEXT-GUIDED GROUP MIXUP WITH CANONICAL MINING FOR IMBALANCED GRAPH CLUSTERING

## ABSTRACT

Graph neural networks (GNNs) have achieved remarkable progress in text-attributed graph clustering. However, these approaches assume that different classes are uniformly distributed, which hinders their applicability in real-world, imbalanced scenarios. Towards this end, this paper studies the problem of imbalanced text-attributed graph clustering, and proposes a novel framework named Text-guided Group Mixup with Canonical Mining (TRACI) for the problem. The core of our TRACI lies in generating mixed groups with an emphasis on minority classes, guided by large language models (LLMs). In particular, we first utilize LLMs to produce diverse views for each sample and randomly assign samples into balanced groups with mixed semantics for consistency learning. To further enhance robustness, we employ LLMs to compute correlation scores among samples with respect to the synthesized groups, thereby reinforcing minority-aware group representations. In addition, we encourage canonical correlations between various augmented views of nodes to ensure semantic alignment. Extensive experiments on several benchmark datasets validate the effectiveness of the proposed TRACI, demonstrating clear advantages over state-of-the-art baselines under class-imbalanced conditions. The source code is available at https://anonymous.4open.science/r/TRACI-E087.

## 1 INTRODUCTION

Graph clustering (Tsitsulin et al., 2023; Ren et al., 2025; Xie et al., 2025), an unsupervised task in graph data mining, aims to assign nodes into distinct clusters that reflect underlying structural and conceptual commonalities. While recent algorithms have made significant progress (Yang et al., 2023; Liu et al., 2023a; 2024b; Kulatilleke et al., 2025), their effectiveness in real-world scenarios remains fundamentally constrained by the inherent class imbalance in graph-structured data (Shi et al., 2020; Huang et al., 2022; Ma et al., 2025). Authentic graph data, such as citation networks (Qin et al., 2025), often exhibit long-tailed distributions (Li & Jia, 2025), where head classes dominate with dense connections, while tail classes are underrepresented, suffering from sparse data and weak connectivity. These naturally occurring imbalances can impair the performance of traditional graph clustering methods, leading to suboptimal results, as such methods are typically developed under the assumption of class balance (Li et al., 2024; Ju et al., 2024; Ma et al., 2025).

To address category imbalance in graph-structured data, researchers have developed three primary paradigms: (i) ***Re-sampling methods*** (Zhang et al., 2023; Gao et al., 2023; Avelino et al., 2024; Carvalho et al., 2025; Nagler et al., 2024) that adjust class selection ratios between classes with varying sample sizes; (ii) ***Re-weighting techniques*** (Li et al., 2025) that modify loss functions based on class frequencies; and (iii) ***Augmentation-based methods*** (Song et al., 2024; Tian et al., 2024; Ding et al., 2025) that transfer knowledge from majority to minority classes using topological or feature semantics. While these approaches have shown promise for attribute graphs with shallow features, they largely neglect the rich contextual semantics in text-attributed graphs (TAGs) (Zhang et al., 2024; He et al., 2025; Hu et al., 2025). This oversight introduces semantic bias, exacerbating challenges such as term frequency-class correlation and topic distribution heterogeneity, which further amplify long-tailed distributions (Chen et al., 2024a). Therefore, effectively leveraging node textual information beyond shallow features, remains a critical challenge in imbalanced text-attributed graph clustering.

The advent of large language models (LLMs) has opened new avenues for text-attributed graph clustering (Chen et al., 2023; Fu et al., 2025). GCLR (Trivedi et al., 2024) leverages the zero-shot capabilities of LLMs to enhance clustering performance through LLM-generated feedback. However, it overlooks the challenge of class imbalance caused by disparities in textual semantics. Although SaVe-TAG (Wang et al., 2024) addresses this issue by synthesizing novel minority-class samples via LLMs for supervised classification, such a strategy is ill-suited for unsupervised clustering and may introduce semantic inconsistencies due to LLM hallucinations (Ji et al., 2023; Verma et al., 2024; Huang et al., 2025). In this work, we seek to harness the zero-shot potential of LLMs for imbalanced graph clustering while striving to preserve semantic consistency in the generated text.

To address the aforementioned challenges, we propose a novel unsupervised LLM-driven framework called Text-guided Group Mixup with Canonical Mining (TRACI) to tackle this problem of imbalance text-attributed graph clustering. The core idea of TRACI is to learn minority-aware group representations that re-balance the contributions of majority and minority classes while preserving semantic integrity as much as possible. We begin by leveraging an LLM to generate augmented texts for nodes in a way that retains their core semantic representations. This is followed by a canonical mining module to align the augmented views in the embedding space. To alleviate imbalance in an unsupervised setting, we randomly assign samples to different groups based on correlation scores provided by the LLM, thereby making better use of textual semantics. For boundary samples between clusters, TRACI utilizes the zero-shot capabilities of LLMs (Ye et al., 2025) to refine the GNN encoder through ranking-based supervision from pseudo-labels generated by the LLM. The effectiveness of TRACI is validated on text-attributed graph datasets and extensive class-imbalanced experiments against state-of-the-art baselines.

In conclusion, our main contributions in this work are summarized as follows:

- **New Perspective.** To the best of our knowledge, we are the first to investigate the problem of imbalanced text-attributed graph clustering enhanced by LLMs in an unsupervised manner.
- **Novel Methodology.** We propose TRACI, a framework that first leverages LLMs to generate text-level augmented views while preserving semantic integrity, followed by the assignment of samples into text-guided mixed groups with canonical correlation alignment.
- **Comprehensive Experiments.** Extensive experiments on multiple benchmark datasets under imbalanced conditions demonstrate that TRACI consistently outperforms state-of-the-art baselines.

## 2 RELATED WORK

**Text-attributed Graph Clustering.** The classic paradigm for text-attributed graph clustering (Tsitsulin et al., 2023; Yan et al., 2023; Zhou et al., 2025; Zhu et al., 2025; Yu et al., 2025) typically involves extracting textual embeddings from shallow, context-free features (Mikolov et al., 2013; Wu et al., 2025; Zhang et al., 2025), which are then integrated with the graph's topological structure via graph neural networks (GNNs) (Liu et al., 2024b; Bhowmick et al., 2024; Wang et al., 2025b). More recently, LLM-based approaches for text-attributed graphs have emerged and can be broadly categorized into three paradigms (Chen et al., 2024b): **LLM-as-Predictor**, **LLM-as-Enhancer** and **LLM-as-Aligner**. Specifically, the **LLM-as-Predictor** paradigm feeds structure-aware textual inputs directly into LLMs to predict node labels (Qiao et al., 2025; Chen et al., 2023). In contrast, **LLM-as-Enhancer** leverages LLMs to enrich text representations, either by extracting contextualized embeddings (fine-tuned (Mavromatis et al., 2023) or frozen (Qiao et al., 2025)) or by generating auxiliary semantic signals (explanations or augmentations). More importantly, **LLM-as-Aligner** aims to align the outputs from GNNs and LLMs iteratively or in parallel (Liu et al., 2025). This paradigm simultaneously leverage the structural aggregation capabilities of GNNs and the semantic extraction abilities of LLMs. These methods are typically implemented through prediction alignment (Zhao et al., 2021) or embedding alignment (Hu et al., 2025). While these paradigms have been actively explored in supervised or self-supervised tasks such as node classification and link prediction, their potential in unsupervised settings like graph clustering remains largely underexplored.

**Long-tailed Graph Learning.** Class imbalance (Carvalho et al., 2025; Ma et al., 2025) in graph data presents a significant obstacle to the effective deployment of Graph Neural Networks (GNNs). Existing efforts to address long-tailed graph learning can be broadly categorized into three main paradigms: re-sampling methods (Carvalho et al., 2025), re-weighting techniques (He, 2024a), and augmentation-based strategies (Khan et al., 2024). On the re-sampling side, GraphSMOTE (Zhao

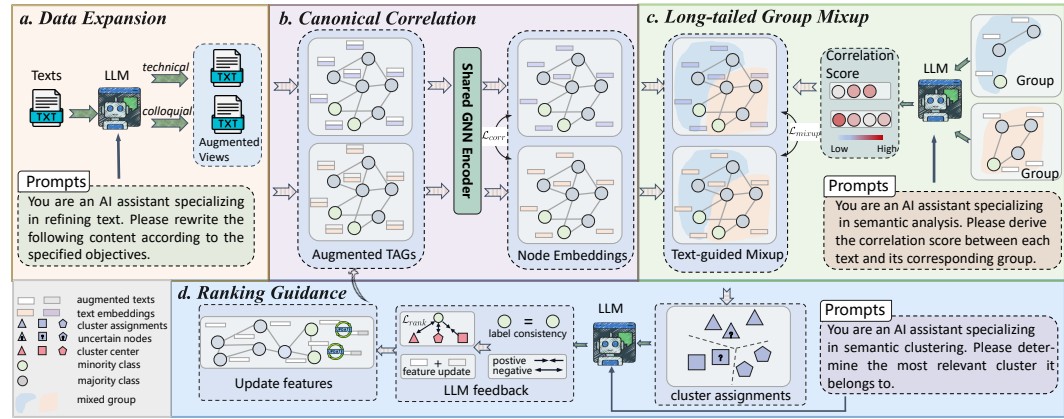

Figure 1: The framework of TRACI. TRACI consists of four key modules: (a) data expansion, (b) canonical correlation, (c) long-tailed group mixup and (d) fine-tuning with ranking guidance.

et al., 2021) and ImGAGN (Qu et al., 2021) generate synthetic samples for minority classes through oversampling and adversarial generation, respectively. On the re-weighting side, ReNode (Chen et al., 2021) adaptively adjusts node weights by quantifying influence shifts near class boundaries. In the augmentation line, RAHNet (Mao et al., 2023) enhances minority class representation through a retrieval-augmented mechanism by incorporating external knowledge. Despite these advancements, most existing methods overlook the rich contextual semantics embedded in node texts (Ghosh et al., 2024; Wang et al., 2025a). To address this, we propose TRACI, a novel framework that leverages textual semantics to generate balanced, minority-aware group representations.

## 3 METHODOLOGY

**Problem Formulation.** A TAG can be represented as $\mathcal{G} = \{\mathcal{V}, \mathbf{A}, \mathcal{D}, \mathbf{X}\}$, where $\mathcal{V}$ denotes a set of $N$ nodes, $\mathbf{A} \in \mathbb{R}^{N \times N}$ is the adjacency matrix, $\mathcal{D} = \{\mathcal{D}_1, \mathcal{D}_2, \cdots, \mathcal{D}_N\}$ represents the text attributes associated with the nodes, and $\mathbf{X} \in \mathbb{R}^{N \times F}$ is the text embedding matrix encoded by the frozen language model Sentence-BERT (Reimers & Gurevych, 2019). In this work, we aim to partition the nodes in $\mathcal{G}$ into $K$ disjoint clusters $\mathcal{C} = \{\mathcal{C}_1, \mathcal{C}_2, \cdots, \mathcal{C}_K\}$ under a long-tailed distribution scenario, where the number of samples in each cluster are highly imbalanced. Specifically, we quantify this skewed distribution using the imbalance ratio, defined as $\rho = \frac{n_{\max}}{n_{\min}}$ (Ma et al., 2025), where $n_{\max} = \max\{|\mathcal{C}k|\}_{k=1}^K$ and $n_{\min} = \min\{|\mathcal{C}k|\}_{k=1}^K$, with $|\cdot|$ denoting the sample size of a set.

### 3.1 FRAMEWORK OVERVIEW

The proposed framework of TRACI is illustrated in Figure 1. The pipeline consists of four key modules: data expansion, canonical correlation, long-tailed mixup and fine-tuning with ranking guidance. Specifically, we leverage an LLM to generate augmented textual views for each node, which are subsequently encoded by the language model (LM) Sentence-BERT to produce input embeddings for the GNN encoder. Subsequently, node-level embeddings are derived from the augmented TAGs using a shared GNN encoder and are aligned in the embedding space via canonical correlation. And the correlation scores computed by the LLM are employed to guide the synthesis of mixed group representations, placing greater emphasis on the minority class and thereby reinforcing minority-aware group representations. Finally, feedback responses from the LLM are utilized to fine-tune TRACI with ranking guidance, enhancing the assignment of boundary nodes.

### 3.2 DATA EXPANSION WITH LARGE LANGUAGE MODELS

In this framework, we follow the standard augmentation-contrastive paradigm, as exemplified by SimCLR (Chen et al., 2020). Previous text augmentation methods (Yan et al., 2021; Gao et al., 2021) typically apply token-level transformations, such as shuffling, dropout, or cutoff to the original text. However, these techniques can inadvertently alter the core semantics of sentences, potentially compromising the performance of downstream tasks. To address this, we forgo such destructive augmentations and instead adopt a more semantically-preserving yet stylistically diverse

strategy powered by LLMs. Specifically, the original text is input into an LLM, which is prompted to generate two stylistically distinct versions: a technical version $\mathcal{D}^{(1)} = \{\mathcal{D}_1^{(1)}, \mathcal{D}_2^{(1)}, \cdots, \mathcal{D}_N^{(1)}\}$ and a colloquial version $\mathcal{D}^{(2)} = \{\mathcal{D}_1^{(2)}, \mathcal{D}_2^{(2)}, \cdots, \mathcal{D}_N^{(2)}\}$. This approach maintains the original semantic content while introducing diverse linguistic expressions, enabling much more abundant semantic representation learning. Finally, we construct two augmented views of the original TAG $\mathcal{G}$: $\mathcal{G}^{(1)} = \{\mathcal{V}, \mathbf{A}, \mathcal{D}^{(1)}, \mathbf{X}^{(1)}\}$ and $\mathcal{G}^{(2)} = \{\mathcal{V}, \mathbf{A}, \mathcal{D}^{(2)}, \mathbf{X}^{(2)}\}$, where $\mathbf{X}^{(1)}$ and $\mathbf{X}^{(2)}$ are embeddings encoded by the augmentations, serving as inputs for following stages.

### 3.3 CANONICAL CORRELATION MAXIMIZATION FOR SEMANTICS ALIGNMENT

To encourage node-level semantic alignment, we adopt a maximum correlation objective (Andrew et al., 2013) to ensure consistency between augmented views. As discussed in Section 3.2, the two augmented views are processed through a shared GNN encoder to promote entity alignment across different stylistic variations. This alignment mechanism ensures that semantically similar entities are mapped closely in the embedding space, regardless of surface-level linguistic differences. Given two augmented views $\mathcal{G}^{(1)}$ and $\mathcal{G}^{(2)}$, we utilize the shared GNN encoder to extract the corresponding node embeddings $\mathbf{Z}^{(1)}$ and $\mathbf{Z}^{(2)} \in \mathbb{R}^{N \times D}$. We further compute the centered node embeddings $\bar{\mathbf{Z}}^{(1)}$ and $\bar{\mathbf{Z}}^{(2)}$ as $\bar{\mathbf{Z}}^{(1)} = \mathbf{Z}^{(1)} - \frac{1}{N}\mathbf{1}_N\mathbf{1}_N^T\mathbf{Z}^{(1)}$ and $\bar{\mathbf{Z}}^{(2)} = \mathbf{Z}^{(2)} - \frac{1}{N}\mathbf{1}_N\mathbf{1}_N^T\mathbf{Z}^{(2)}$, respectively. Here, $\mathbf{1}_N$ is a vector length of $N$ with all elements equal to 1. The cross-covariance matrix between the two views is calcultaed as $\mathbf{C}_{1,2} = (\bar{\mathbf{Z}}^{(1)})^\top\bar{\mathbf{Z}}^{(2)}/(N-1) \in \mathbb{R}^{D \times D}$, and the self-covariance matrix for each view is given by $\mathbf{C}_{1,1} = (\bar{\mathbf{Z}}^{(1)})^\top\bar{\mathbf{Z}}^{(1)}/(N-1)$. Finally, the canonical correlation loss between the augmented views is applied to align them in the embedding space, which is defined as:

$$\mathcal{L}_{\text{corr}}(\mathbf{Z}^{(1)}, \mathbf{Z}^{(2)}) = -\text{Trace}(\mathbf{C}_{1,1}^{-1/2}\mathbf{C}_{1,2}\mathbf{C}_{2,2}^{-1}\mathbf{C}_{2,1}\mathbf{C}_{2,2}^{-1/2}). \tag{1}$$

### 3.4 LONG-TAILED GROUP MIXUP WITH TEXTUAL GUIDANCE

To alleviate the imbalance issue, we design a mixup strategy guided by textual semantics to learn minority-aware group representations. Specifically, samples from the augmented view $\mathcal{G}^{(1)}$ are randomly partitioned into $M$ groups $\mathbb{G} = \{\mathbb{G}_1, \mathbb{G}_2, \cdots, \mathbb{G}_M\}$, where $\mathbb{G}_m$ represents the index set of samples in the $m$-th group. The corresponding samples from the other augmented view $\mathcal{G}^{(2)}$ are partitioned with the same index sets. Subsequently, each group of texts is fed into an LLM using the prompt $\mathcal{P}_{\text{mix}}$, which outputs a contribution score $b_{mn}^{(i)}$ and a confidence score $c_{mn}^{(i)}$ for the $n$-th text in the $m$-th group from the $i$-th augmented view. The contribution score $b_{mn}^{(i)}$ reflects the semantic relevance and conceptual coherence of the text within the group, while the confidence score $c_{mn}^{(i)}$ estimates the credibility of the corresponding contribution. Based on these scores, we finally derive a correlation-based weight matrix $\mathbf{S}^{(i)} \in \mathbb{R}^{M \times N}$ for each view ($i = 1, 2$) to improve awareness of minority in the mixed groups, whose element-wise definitions are stated as follows:

$$s_{mn}^{(i)} = \frac{e^{(1-b_{mn}^{(i)})\cdot c_{mn}^{(i)}}}{\sum\limits_{n \in \mathbb{G}_m} e^{(1-b_{mn}^{(i)})\cdot c_{mn}^{(i)}}}\mathbf{1}_{n \in \mathbb{G}_m}. \tag{2}$$

Here, $\mathbf{1}_{n \in \mathbb{G}_m}$ is an indicator function that equals 1 if and only if $n \in \mathbb{G}_m$. Subsequently, we construct group-level synthetic embeddings as weighted combinations of the sample representations,

$$\mathbf{h}_m^{(i)} = \sum_{n \in \mathbb{G}_m} s_{mn}^{(i)}\mathbf{z}_n^{(i)} / \|\sum_{n \in \mathbb{G}_m} s_{mn}^{(i)}\mathbf{z}_n^{(i)}\|_2, \tag{3}$$

where $\|\cdot\|_2$ denotes the $\ell_2$ norm and $\mathbf{z}_n^{(i)}$ is the representation of the $n$-th sample in the $i$-th view. Following the aforementioned steps, we obtain two group-level augmented representations $\mathbf{H}^{(1)}$ and $\mathbf{H}^{(2)} \in \mathbb{R}^{M \times D}$ for contrastive learning. In the contrastive learning setup, corresponding group representations from the two views form positive pairs, while all other combinations are considered negative pairs. The imbalance-aware contrastive loss is then defined as:

$$\mathcal{L}_{\text{mixup}}(\mathbf{H}^{(1)}, \mathbf{H}^{(2)}) = -\frac{1}{M}\sum_{m=1}^{M}\log\frac{e^{\theta(\mathbf{h}_m^{(1)}, \mathbf{h}_m^{(2)})/\tau_1}}{e^{\theta(\mathbf{h}_m^{(1)}, \mathbf{h}_m^{(2)})/\tau_1} + \sum\limits_{m' \neq m} e^{\theta(\mathbf{h}_m^{(1)}, \mathbf{h}_{m'}^{(1)})/\tau_1} + \sum\limits_{m' \neq m} e^{\theta(\mathbf{h}_m^{(1)}, \mathbf{h}_{m'}^{(2)})/\tau_1}},$$
$$\tag{4}$$

where $\tau_1$ is a temperature hyperparameter and $\theta$ denotes cosine similarity between two vectors.

## 3.5 FINE-TUNING WITH RANKING GUIDANCE

To further leverage the capabilities of LLMs to enhance the GNN encoder, we propose a two-stage optimization strategy for TRACI. In the first stage, we align two augmented views in the embedding space while applying text-guided mixup to learn minority-aware group representations. In the second stage, we utilize LLMs to annotate the boundaries of imbalanced nodes, particularly between clusters that are difficult for the GNN to distinguish. The feedback obtained from the LLM is then used as ranking guidance to fine-tune the encoder for better representation learning. To jointly enforce consistency between both augmented views and address the problem of class imbalance, we combine Equation 1 and Equation 4 as follows, where $\alpha$ denotes the trade-off hyperparameter:

$$\mathcal{L}_{\text{warm}} = \alpha \mathcal{L}_{\text{corr}} + (1 - \alpha)\mathcal{L}_{\text{mixup}}. \tag{5}$$

**Boundary Nodes for Querying LLMs.** After completing the first-stage training, we obtain the embeddings $\mathbf{Z}^{(1)}$ and $\mathbf{Z}^{(2)}$ of the two augmented views using the frozen encoder $f_\Theta$, expressed as,

$$\mathbf{Z}^{(1)} = f_\Theta(\mathbf{A}, \mathbf{X}^{(1)}) \text{ and } \mathbf{Z}^{(2)} = f_\Theta(\mathbf{A}, \mathbf{X}^{(2)}), \tag{6}$$

where $\Theta$ is the learned parameter. To mitigate uniform clustering under imbalanced settings, we apply smooth $k$-means clustering (He, 2024b) to these embeddings, yielding the predicted label sets $\mathcal{C}^{(1)}$ and $\mathcal{C}^{(2)}$, respectively. We then identify a set of challenging nodes $\mathcal{S} = \{i \in \mathcal{V} \mid \mathcal{C}_i^{(1)} \neq \mathcal{C}_i^{(2)}\}$, which are subsequently used as queries to the LLM to obtain additional guidance for better learning.

**Concept Induction for Each Cluster.** Since the semantic meanings of clusters are initially unknown, we select the top-$k$ nearest nodes to each cluster centroid and construct a representative sample set to query the LLM for inducing core concepts of each cluster via the prompt $\mathcal{P}_{\text{indu}}$, expressed as follows:

$$\mathcal{M} = \mathcal{P}_{\text{indu}}(\text{top-}k \text{ texts for each cluster}). \tag{7}$$

**Decision Filtering with Ranking Guidance.** Subsequently, we feed the derived concept set $\mathcal{M}$ together with the textual content of the challenging node set $\mathcal{S}$ from both augmented views into the LLM using the prompt $\mathcal{P}$pred, yielding the predicted label sets $\mathcal{C}_{\text{chal}}^{(1)}$ and $\mathcal{C}_{\text{chal}}^{(2)}$, as follows:

$$\mathcal{C}_{\text{chal}}^{(1)} = \mathcal{P}_{\text{pred}}(\mathcal{M}, \mathcal{D}_S^{(1)}), \ \mathcal{C}_{\text{chal}}^{(2)} = \mathcal{P}_{\text{pred}}(\mathcal{M}, \mathcal{D}_S^{(2)}). \tag{8}$$

Given that LLM predictions may inevitably contain errors, particularly for challenging nodes, we introduce a dual-view consensus mechanism to filter out noisy labels and mitigate additional biases introduced by the LLM. The detailed filtering process is described as follows:

$$\mathcal{C}_{\text{LLM}} = \{i \in \mathcal{S} \mid \mathcal{C}_{\text{chal}}^{(1)}(i) = \mathcal{C}_{\text{chal}}^{(2)}(i)\}. \tag{9}$$

More importantly, we further apply a ranking-based contrastive loss to incorporate the LLM's feedback and fine-tune the GNN model obtained in the first stage. Specifically,

$$\mathcal{L}_{\text{rank}} = -\sum_{i \in \mathcal{C}_{\text{LLM}}} \log \frac{\exp\left(\theta\left(\mathbf{z}_i, \boldsymbol{\mu}_{\mathcal{C}_{\text{chal}}^{(1)}(i)}\right)/\tau_2\right)}{\sum_{k=1}^{K} \exp\left(\theta\left(\mathbf{z}_i, \boldsymbol{\mu}_k\right)/\tau_2\right)}. \tag{10}$$

Here, $\tau_2$ denotes the temperature hyper-parameter, and $\theta$ represents the cosine similarity between the two vectors. $\boldsymbol{\mu}_k$ refers to the cluster centroid of the $k$th cluster. We then employ the objective $\mathcal{L}_{\text{fine}}$ to fine-tune the final imbalanced graph clustering model:

$$\mathcal{L}_{\text{fine}} = \alpha \mathcal{L}_{\text{corr}} + (1 - \alpha)(\beta \mathcal{L}_{\text{mixup}} + (1 - \beta)\mathcal{L}_{\text{rank}}). \tag{11}$$

Consequently, we first warm up a base model capable of handling imbalanced representation learning across two augmented views. Building upon this foundation, we identify challenging nodes and leverage LLMs to further enhance the model's ability to address class imbalance. This progressive paradigm of TRACI results in more reliable predictions for imbalanced graph clustering.

### 3.6 THEORETICAL ANALYSIS

Theoretically, our theorem establishes a tighter generalization error bound with the re-balanced mixup strategy guided by the LLM, compared to the standard contrastive loss.

**Theorem 3.1.** *Let $\mathcal{X}$ be the input space and $\mathcal{Z} \subset \mathbb{R}^D$ denotes the latent space. Suppose the following conditions holds: (1) **Imbalanced Distribution**: $\exists \rho \gg 1$ s.t. $n_{max}/n_{min} = \rho$ for cluster sizes; (2) **Group Mixup**: Samples partitioned into $M$ groups $\{\mathbb{G}_m\}_{m=1}^M$ such that $||\mathbb{G}_m|| = n/M$. (3) **LLM-guided Weights**: The weight matrix $\mathbf{S} \in \mathbb{R}^{M \times N}$ in Eq. (2) satisfying $\sum_n s_{m,n} = 1$. Then, the generalization error bound for the mixup loss $\mathcal{L}_{mixup}$ is tighter than that of the classic contrastive loss $\mathcal{L}_{cl}$. Specifically, with probability at least $1 - \delta$, for any encoder $f_\Theta$, the following holds:*

$$\mathcal{E}(\mathcal{L}_{mixup}) - \mathcal{E}^*(\mathcal{L}_{mixup}) \leq C(\sqrt{\frac{\log M}{M}} + \sqrt{\frac{\log(1/\delta)}{N}}), \tag{12}$$

$$\mathcal{E}(\mathcal{L}_{cl}) - \mathcal{E}^*(\mathcal{L}_{cl}) \leq C(\sqrt{\frac{\log N}{N}} + \sqrt{\frac{\log(1/\delta)}{N}}), \tag{13}$$

*where $C > 0$ is a constant, $M \ll N$, $\mathcal{E}$ denotes the generalization error, and $\mathcal{E}^*$ is the Bayes optimal error. $\mathcal{L}_{cl}$ is the classic contrastive loss in the embedding space.*

The proof sketch consists of three key steps: (1) reducing the effective sample complexity through group-wise mixup, (2) bounding the empirical Rademacher complexity for both of $\mathcal{L}_{mixup}$ and $\mathcal{L}_{cl}$, and (3) incorporating statistical learning theory to establish generalization bounds. A detailed proof of Theorem 3.1 can be found in the Appendix, which provides theoretical support for our TRACI.

## 4 EXPERIMENTS

### 4.1 EXPERIMENTAL SETUP

**Imbalanced Datasets.** In this work, we evaluate the performance of TRACI under class-imbalanced scenarios using four widely adopted TAG datasets: Cora (McCallum et al., 2000), CiteSeer (Giles et al., 1998), WikiCS (Mernyei & Cangea, 2020), and PubMed (Sen et al., 2008). To simulate real-world imbalance, we construct long-tailed variants of these datasets (Park et al., 2021) with varying imbalance ratios. Specifically, the sample size for the $k$-th class is given by $n_k = n_{max} \cdot \rho^{-\frac{k-1}{K-1}}$ (Ma et al., 2025), with $K$ denoting the total number of classes. To preserve the topological structure of the original graph, nodes with higher connectivity are preferentially retained during the sampling process. Detailed statistics corresponding to different imbalance ratios ($\rho = 10, 20, 50, 100$) are illustrated in the following Figure 3. Additional details about the datasets, including their topological structures, are provided comprehensively in Table A.

**Baseline Methods.** We compare TRACI with several state-of-the-art deep clustering methods: DMoN (Tsitsulin et al., 2023), Dink-Net (Liu et al., 2023a), HSAN (Liu et al., 2023b), S$^3$GC (Devvrit et al., 2022), DGCLUSTER (Bhowmick et al., 2024), MAGI (Liu et al., 2024b) and IsoSEL (Sun et al., 2025), under class-imbalanced settings. In particular, we extend our evaluation by comparing TRACI with established graph learning frameworks such as GraphSMOTE (Zhao et al., 2021), GraphENS (Park et al., 2021), and BAT (Liu et al., 2024c), thereby providing a more comprehensive validation of its effectiveness. Following prior work, we adopt accuracy (ACC), normalized mutual information (NMI), and F1 score as evaluation metrics for comparison.

**Implementation Details.** The implementation of our proposed method, TRACI, is based on the PyTorch library, and both the datasets and source codes are publicly available. To encode textual information, we utilize Sentence-BERT (Reimers & Gurevych, 2019) to extract text embeddings. For representation learning, we employ a Graph Convolutional Network (GCN) as the backbone encoder for GNN to aggregate neighborhood semantics. For interactions with large language models (LLMs), we use ChatGPT (gpt-4o-mini) (Hurst et al., 2024) to provide guidance and feedback; the detailed prompt design is described in Table E. For fair comparison, we report the performance as the mean and standard deviation over five runs. In particular, more detailed information, including hyperparameter settings and the training strategy, is thoroughly provided in Table H.

Table 1: Clustering performance of TRACI compared to baseline methods under varying imbalance ratios ($\rho = 10$ and $20$). Boldfaced scores indicate the best results, while underlined scores denote the second-best results. "OOM" means out-of-memory.

| $\rho$ | Dataset | Metric | DMoN | Dink-Net | HSAN | S³GC | DGCluster | MAGI | IsoSEL | TRACI |
|---|---|---|---|---|---|---|---|---|---|---|
| 10 | Cora | ACC | $60.05_{\pm3.08}$ | $61.67_{\pm1.34}$ | $63.41_{\pm2.17}$ | $56.68_{\pm2.64}$ | $56.86_{\pm4.19}$ | $\underline{65.34}_{\pm0.31}$ | $58.66_{\pm6.07}$ | $\mathbf{73.48}_{\pm2.21}$ |
| | | NMI | $49.61_{\pm0.72}$ | $51.88_{\pm0.95}$ | $49.26_{\pm1.05}$ | $43.54_{\pm0.51}$ | $52.80_{\pm0.94}$ | $\underline{52.90}_{\pm0.30}$ | $50.32_{\pm1.70}$ | $\mathbf{55.60}_{\pm0.10}$ |
| | | F1 | $51.02_{\pm2.90}$ | $55.57_{\pm4.24}$ | $57.88_{\pm1.55}$ | $51.26_{\pm2.80}$ | $49.90_{\pm5.59}$ | $\underline{60.04}_{\pm0.26}$ | $43.70_{\pm7.19}$ | $\mathbf{67.03}_{\pm3.79}$ |
| | CiteSeer | ACC | $56.97_{\pm8.96}$ | $55.88_{\pm4.77}$ | $55.28_{\pm2.29}$ | $56.63_{\pm2.17}$ | $47.48_{\pm3.37}$ | $\underline{63.13}_{\pm0.34}$ | $54.73_{\pm12.43}$ | $\mathbf{67.15}_{\pm0.17}$ |
| | | NMI | $39.64_{\pm3.42}$ | $37.63_{\pm0.56}$ | $37.31_{\pm0.58}$ | $40.10_{\pm0.56}$ | $36.62_{\pm0.83}$ | $\underline{40.75}_{\pm0.40}$ | $37.64_{\pm1.15}$ | $\mathbf{41.72}_{\pm0.10}$ |
| | | F1 | $49.17_{\pm10.68}$ | $46.28_{\pm4.93}$ | $50.56_{\pm3.97}$ | $51.59_{\pm1.19}$ | $32.09_{\pm4.11}$ | $\underline{56.44}_{\pm0.38}$ | $37.39_{\pm10.31}$ | $\mathbf{60.44}_{\pm0.14}$ |
| | WikiCS | ACC | $34.73_{\pm1.08}$ | $54.89_{\pm3.57}$ | $56.34_{\pm3.55}$ | $44.64_{\pm2.21}$ | $55.40_{\pm2.36}$ | $\underline{58.55}_{\pm0.67}$ | | $\mathbf{63.33}_{\pm1.55}$ |
| | | NMI | $24.14_{\pm1.41}$ | $44.30_{\pm1.65}$ | $48.28_{\pm1.06}$ | $38.37_{\pm0.47}$ | $43.93_{\pm1.57}$ | $\mathbf{49.58}_{\pm0.15}$ | OOM | $\underline{49.23}_{\pm1.25}$ |
| | | F1 | $27.30_{\pm2.12}$ | $46.60_{\pm6.97}$ | $\underline{50.76}_{\pm3.88}$ | $36.48_{\pm2.58}$ | $46.59_{\pm5.44}$ | $47.47_{\pm0.56}$ | | $\mathbf{53.93}_{\pm1.60}$ |
| | PubMed | ACC | $55.62_{\pm8.84}$ | $49.11_{\pm3.90}$ | $49.26_{\pm0.03}$ | $54.07_{\pm0.75}$ | $\underline{58.31}_{\pm2.96}$ | $41.79_{\pm0.15}$ | $58.24_{\pm5.59}$ | $\mathbf{61.42}_{\pm5.30}$ |
| | | NMI | $8.17_{\pm1.46}$ | $11.01_{\pm2.02}$ | $7.95_{\pm0.03}$ | $\underline{15.16}_{\pm0.99}$ | $10.98_{\pm1.00}$ | $8.08_{\pm0.03}$ | $12.69_{\pm3.39}$ | $\mathbf{20.90}_{\pm1.61}$ |
| | | F1 | $42.91_{\pm3.81}$ | $41.88_{\pm5.26}$ | $42.47_{\pm0.03}$ | $\underline{47.58}_{\pm0.90}$ | $47.17_{\pm4.01}$ | $29.86_{\pm0.11}$ | $42.38_{\pm5.18}$ | $\mathbf{52.45}_{\pm4.98}$ |
| 20 | Cora | ACC | $60.17_{\pm5.01}$ | $58.20_{\pm2.39}$ | $53.38_{\pm1.81}$ | $51.30_{\pm0.88}$ | $53.18_{\pm2.08}$ | $57.86_{\pm0.77}$ | $\underline{60.28}_{\pm9.95}$ | $\mathbf{68.89}_{\pm6.08}$ |
| | | NMI | $\underline{50.52}_{\pm1.01}$ | $47.81_{\pm1.16}$ | $46.99_{\pm0.80}$ | $43.52_{\pm1.45}$ | $48.71_{\pm0.27}$ | $49.33_{\pm0.33}$ | $48.88_{\pm1.41}$ | $\mathbf{52.56}_{\pm3.17}$ |
| | | F1 | $46.05_{\pm3.91}$ | $46.05_{\pm3.36}$ | $47.22_{\pm2.03}$ | $43.10_{\pm1.17}$ | $42.97_{\pm1.23}$ | $\underline{51.68}_{\pm0.49}$ | $41.68_{\pm11.24}$ | $\mathbf{57.65}_{\pm4.92}$ |
| | CiteSeer | ACC | $\underline{59.07}_{\pm4.98}$ | $50.87_{\pm3.30}$ | $49.12_{\pm0.67}$ | $53.58_{\pm1.01}$ | $47.93_{\pm3.01}$ | $58.92_{\pm0.91}$ | $50.41_{\pm8.28}$ | $\mathbf{67.15}_{\pm6.50}$ |
| | | NMI | $39.67_{\pm1.59}$ | $37.56_{\pm1.07}$ | $31.96_{\pm1.03}$ | $\underline{40.67}_{\pm1.31}$ | $36.22_{\pm1.11}$ | $37.80_{\pm0.28}$ | $31.50_{\pm4.03}$ | $\mathbf{42.59}_{\pm2.43}$ |
| | | F1 | $46.46_{\pm5.56}$ | $39.45_{\pm4.80}$ | $43.11_{\pm1.88}$ | $44.31_{\pm1.54}$ | $27.58_{\pm5.13}$ | $\underline{50.33}_{\pm0.64}$ | $31.02_{\pm2.96}$ | $\mathbf{55.67}_{\pm7.68}$ |
| | WikiCS | ACC | $37.29_{\pm0.19}$ | $58.73_{\pm5.74}$ | $55.61_{\pm5.75}$ | $45.41_{\pm0.28}$ | $59.28_{\pm0.78}$ | $\underline{60.01}_{\pm0.07}$ | | $\mathbf{60.57}_{\pm3.37}$ |
| | | NMI | $28.22_{\pm0.85}$ | $\underline{48.48}_{\pm2.16}$ | $47.40_{\pm2.02}$ | $40.00_{\pm0.17}$ | $46.71_{\pm0.71}$ | $\mathbf{49.75}_{\pm0.09}$ | OOM | $47.40_{\pm0.98}$ |
| | | F1 | $27.61_{\pm2.04}$ | $\mathbf{48.94}_{\pm6.62}$ | $45.41_{\pm4.03}$ | $35.76_{\pm0.19}$ | $44.69_{\pm2.45}$ | $45.88_{\pm0.07}$ | | $\underline{48.59}_{\pm1.07}$ |
| | PubMed | ACC | $56.01_{\pm11.16}$ | $47.77_{\pm3.25}$ | $44.35_{\pm3.33}$ | $50.65_{\pm0.49}$ | $57.34_{\pm3.23}$ | $45.60_{\pm0.16}$ | $\underline{59.31}_{\pm5.70}$ | $\mathbf{64.72}_{\pm5.28}$ |
| | | NMI | $7.34_{\pm1.61}$ | $7.49_{\pm0.29}$ | $4.92_{\pm0.28}$ | $\underline{12.12}_{\pm0.41}$ | $8.24_{\pm0.45}$ | $6.79_{\pm0.09}$ | $8.03_{\pm0.87}$ | $\mathbf{13.76}_{\pm1.35}$ |
| | | F1 | $\underline{40.59}_{\pm5.52}$ | $34.12_{\pm5.48}$ | $32.70_{\pm4.19}$ | $40.58_{\pm0.52}$ | $38.62_{\pm5.11}$ | $29.29_{\pm0.04}$ | $36.08_{\pm3.80}$ | $\mathbf{49.43}_{\pm3.11}$ |

## 4.2 Performance Comparison

**Performance of graph clustering under imbalance.** To comprehensively assess the performance of TRACI, we evaluate it against seven state-of-the-art baseline methods under imbalance ratios of 10 and 20. As shown in Table 1, TRACI consistently outperforms other state-of-the-art methods on the Cora, CiteSeer, and PubMed datasets across all three metrics (ACC, NMI, and F1 score) under both imbalance ratios ($\rho = 10$ and $\rho = 20$). Specifically, on the Cora dataset with $\rho = 10$, TRACI achieves improvements of 8.14%, 2.70%, and 6.99% in ACC, NMI, and F1 score, respectively, compared to the second-best baseline MAGI. On the WikiCS dataset, TRACI achieves the highest accuracy scores under both $\rho = 10$ and $\rho = 20$. Although the NMI and F1 scores are not the highest in all cases, they still rank among the top-performing methods. Interestingly, some baseline methods demonstrate high ACC but relatively low NMI (e.g., DGCluster on PubMed under $\rho = 10$). This discrepancy may be attributed to misclustered samples: either large clusters are fragmented into smaller ones, or small clusters are merged into a larger one, which distorts the clustering quality despite a seemingly good accuracy. In contrast, TRACI produces clustering results that more faithfully reflect the underlying long-tailed distribution, making it particularly well-suited for real-world class-imbalanced scenarios. Overall, these findings strongly affirm the effectiveness and robustness of TRACI in handling imbalanced data. Furthermore, we evaluate TRACI under even more severe imbalance conditions ($\rho = 50$ and $100$), and the corresponding results for more imbalanced scenarios are provided comprehensively in Table 9.

**Performance of graph learning under imbalance.** We evaluate the representations learned by TRACI against other imbalanced graph learning methods on our established datasets. Specifically, we use the learned representations with an additional classifier for node classification. As shown in Table 2, although TRACI performs relatively poorly on Cora, it consistently achieves superior performance on CiteSeer, WikiCS, and PubMed, demonstrating

Table 2: Graph learning Performance of TRACI in comparison with baselines.

| Dataset | Metric | GraphSMOTE | GraphENS | BAT | TRACI |
|---|---|---|---|---|---|
| Cora | ACC | $83.47_{\pm1.28}$ | $84.00_{\pm1.30}$ | $\underline{84.84}_{\pm0.95}$ | $\mathbf{84.92}_{\pm0.97}$ |
| | NMI | $65.31_{\pm2.15}$ | $65.71_{\pm3.30}$ | $\mathbf{67.31}_{\pm1.39}$ | $\underline{65.90}_{\pm1.96}$ |
| | F1 | $77.75_{\pm1.89}$ | $79.56_{\pm2.50}$ | $\mathbf{80.76}_{\pm1.30}$ | $\underline{80.47}_{\pm1.53}$ |
| CiteSeer | ACC | $73.31_{\pm1.03}$ | $73.14_{\pm2.56}$ | $\underline{75.54}_{\pm0.56}$ | $\mathbf{78.91}_{\pm2.44}$ |
| | NMI | $46.07_{\pm1.70}$ | $46.11_{\pm2.85}$ | $\underline{48.79}_{\pm1.17}$ | $\mathbf{52.41}_{\pm5.12}$ |
| | F1 | $64.86_{\pm1.40}$ | $64.25_{\pm2.32}$ | $\underline{66.35}_{\pm1.23}$ | $\mathbf{66.88}_{\pm4.37}$ |
| WikiCS | ACC | $81.32_{\pm1.54}$ | $80.28_{\pm0.97}$ | $\underline{81.56}_{\pm0.54}$ | $\mathbf{82.01}_{\pm0.93}$ |
| | NMI | $62.83_{\pm2.21}$ | $61.52_{\pm1.49}$ | $\underline{63.12}_{\pm0.76}$ | $\mathbf{63.35}_{\pm1.36}$ |
| | F1 | $78.62_{\pm1.76}$ | $77.52_{\pm0.56}$ | $\underline{79.07}_{\pm0.53}$ | $\mathbf{79.29}_{\pm0.70}$ |
| PubMed | ACC | $86.40_{\pm1.10}$ | $84.72_{\pm0.39}$ | $\underline{87.14}_{\pm0.37}$ | $\mathbf{89.23}_{\pm0.91}$ |
| | NMI | $45.02_{\pm1.92}$ | $42.47_{\pm0.96}$ | $\underline{45.64}_{\pm0.84}$ | $\mathbf{50.23}_{\pm3.00}$ |
| | F1 | $76.82_{\pm1.38}$ | $75.06_{\pm0.45}$ | $\underline{77.14}_{\pm0.57}$ | $\mathbf{79.23}_{\pm0.96}$ |

Table 3: Comparison of TRACI with various model variants in terms of ACC and F1 scores under imbalanced settings ($\rho = 10$ and 20). The best results are highlighted in bold.

| Imbalance | Variants | Cora | | CiteSeer | | WikiCS | | PubMed | |
|---|---|---|---|---|---|---|---|---|---|
| | | ACC | F1 | ACC | F1 | ACC | F1 | ACC | F1 |
| $\rho = 10$ | w/o LLM Expansion | $69.45_{\pm2.15}$ | $61.83_{\pm1.77}$ | $64.96_{\pm3.69}$ | $55.50_{\pm7.08}$ | $51.16_{\pm2.96}$ | $46.49_{\pm1.32}$ | $57.37_{\pm0.15}$ | $47.83_{\pm0.07}$ |
| | w/o LLM Mixup | $70.19_{\pm2.60}$ | $61.29_{\pm3.51}$ | $64.31_{\pm6.59}$ | $54.81_{\pm8.35}$ | $61.33_{\pm0.34}$ | $52.04_{\pm0.65}$ | $59.12_{\pm6.12}$ | $50.16_{\pm5.87}$ |
| | w/o Smooth | $68.60_{\pm4.17}$ | $63.11_{\pm3.69}$ | $60.06_{\pm5.14}$ | $51.36_{\pm6.54}$ | $58.23_{\pm3.04}$ | $53.13_{\pm1.67}$ | $59.72_{\pm5.56}$ | $51.13_{\pm4.90}$ |
| | **TRACI** | $\mathbf{73.48}_{\pm2.21}$ | $\mathbf{67.03}_{\pm3.79}$ | $\mathbf{67.15}_{\pm0.17}$ | $\mathbf{60.44}_{\pm0.14}$ | $\mathbf{63.33}_{\pm1.55}$ | $\mathbf{53.93}_{\pm1.60}$ | $\mathbf{61.42}_{\pm5.30}$ | $\mathbf{52.45}_{\pm4.98}$ |
| $\rho = 20$ | w/o LLM Expansion | $66.12_{\pm3.63}$ | $57.35_{\pm3.54}$ | $61.63_{\pm7.44}$ | $51.78_{\pm7.78}$ | $50.92_{\pm4.35}$ | $43.57_{\pm2.53}$ | $50.29_{\pm0.30}$ | $39.40_{\pm0.19}$ |
| | w/o LLM Mixup | $61.77_{\pm3.19}$ | $52.79_{\pm0.84}$ | $64.08_{\pm6.73}$ | $52.11_{\pm5.42}$ | $53.83_{\pm3.55}$ | $45.30_{\pm1.89}$ | $60.44_{\pm0.13}$ | $47.38_{\pm0.08}$ |
| | w/o Smooth | $60.09_{\pm4.16}$ | $49.96_{\pm5.93}$ | $58.35_{\pm4.36}$ | $49.04_{\pm3.13}$ | $54.96_{\pm4.26}$ | $44.58_{\pm1.39}$ | $56.38_{\pm0.22}$ | $45.34_{\pm0.15}$ |
| | **TRACI** | $\mathbf{68.89}_{\pm6.08}$ | $\mathbf{57.65}_{\pm4.92}$ | $\mathbf{67.15}_{\pm6.50}$ | $\mathbf{55.67}_{\pm7.68}$ | $\mathbf{60.57}_{\pm3.37}$ | $\mathbf{48.59}_{\pm1.07}$ | $\mathbf{64.72}_{\pm5.28}$ | $\mathbf{49.43}_{\pm3.11}$ |

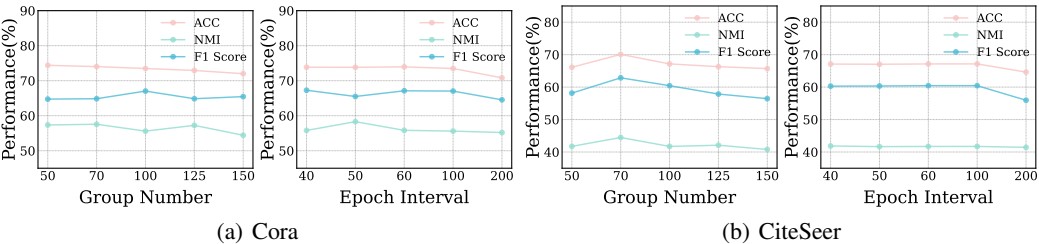

(a) Cora        (b) CiteSeer

Figure 2: Sensitivity analysis of group number and epoch interval for updating correlation scores under an imbalance ratio of 10.

both the effectiveness and robustness of the proposed approach. More comprehensively, these results further validate that TRACI enhances representation learning in the hidden space, leading to more discriminative representations.

## 4.3 ABLATION STUDY

To further investigate the contribution of each module in TRACI, we introduce three variant models to evaluate the effectiveness of individual components, described as follows: (1) **TRACI w/o LLM Expansion**: This variant removes the text-level augmentation generated by LLMs and replaces it with random perturbations to node-level features. (2) **TRACI w/o LLM Mixup**: This version omits the computation of correlation scores using LLMs and instead uses the average of sample embeddings to generate group representations. (3) **TRACI w/o Smooth**: This variant employs hard k-means clustering in the embedding space, replacing smooth k-means which is designed to mitigate the effects of class imbalance. Based on above variants, the effectiveness of TRACI can be demonstrated.

Table 3 presents the performance of TRACI and its variants on the Cora, CiteSeer, WikiCS, and PubMed datasets under imbalance ratios $\rho = 10$ and $\rho = 20$. Several key observations can be drawn from the results: First, **TRACI w/o LLM Expansion** exhibits a significant performance decline compared to the full model TRACI, indicating that the diverse views generated by LLMs provide richer semantic information at both the textual and embedding levels than simple feature perturbations. Second, the performance of **TRACI w/o LLM Mixup** also deteriorates under imbalanced settings, highlighting the importance of learning minority-aware group representations using correlation scores derived from LLMs based on semantic relevance. Finally, removing smooth k-means in **TRACI w/o Smooth** leads to further performance drops, demonstrating its effectiveness in mitigating overly uniform clustering and better handling real-world imbalanced conditions. Overall, these ablation results comprehensively validate the contributions of each module in enhancing the robustness and performance of TRACI.

## 4.4 SENSITIVITY ANALYSIS ON HYPERPARAMETERS AND LLM CHOICE

In this section, we conduct a sensitivity analysis of TRACI from two perspectives: (i) the impact of hyperparameter configurations on model performance, and (ii) the effect of different LLM choices on the overall effectiveness of TRACI. The results are presented in Figure 2 and Table 4.

**Effect of Hyperparameters.** We examine two hyperparameters involved in the text-guided mixup process: the number of groups and the epoch interval for updating correlation scores. By default, the group number and the epoch interval are set to 100 and 50, respectively, for both the Cora and

CiteSeer datasets, as reported in Table 1. In the sensitivity analysis, we vary the group number in {50, 75, 100, 125 150 } and the epoch interval in {40, 50, 60, 100, 200}, while keeping all other hyperparameters fixed. Figure 2 presents the ACC, NMI and F1 scores under an imbalance ratio of 10. On the Cora dataset, the performance generally declines as the group number increases (e.g., ACC drops from 74.39% to 72.00%) and as the epoch interval becomes longer (e.g., ACC drops from 73.85% to 70.82%). This observations suggest that an excessively large number of groups may impair the model's ability to capture minority-aware representations, while a prolonged epoch interval may result in insufficient updates to the mixup groups, thereby weakening the interaction between majority and minority classes in a long-tailed distribution. In conclusion, although the sensitivity trends differ slightly across datasets under an imbalance ratio of 10, TRACI consistently demonstrates robust and competitive performance across a wide range of hyperparameter settings.

**Effect of LLM Selection.** In our TRACI, we leverage the capability of large language models (LLMs) to generate text-level augmented views and to comprehend contextual semantics, thereby enhancing performance in imbalance graph clustering. To quantitatively assess the impact of

Table 4: Effect of LLM selection on TRACI' performance.

| Dataset | Metric | DeepSeek-V3 | GPT-3.5 | GPT-4o-mini | GPT-4.1-mini |
|---------|--------|-------------|---------|-------------|--------------|
| Cora | ACC | $69.27_{\pm 0.39}$ | $73.16_{\pm 1.05}$ | $73.48_{\pm 2.21}$ | $70.07_{\pm 4.77}$ |
| | NMI | $53.81_{\pm 0.39}$ | $56.52_{\pm 1.64}$ | $55.60_{\pm 0.10}$ | $55.22_{\pm 1.83}$ |
| | F1 | $61.41_{\pm 0.43}$ | $63.79_{\pm 0.77}$ | $67.03_{\pm 3.79}$ | $61.11_{\pm 6.13}$ |
| CiteSeer | ACC | $66.74_{\pm 7.13}$ | $68.92_{\pm 0.83}$ | $67.15_{\pm 0.17}$ | $68.14_{\pm 0.03}$ |
| | NMI | $42.95_{\pm 2.37}$ | $43.68_{\pm 0.20}$ | $41.72_{\pm 0.10}$ | $43.40_{\pm 0.00}$ |
| | F1 | $58.75_{\pm 7.26}$ | $61.34_{\pm 0.99}$ | $60.44_{\pm 0.14}$ | $61.77_{\pm 0.01}$ |

LLM selection, we evaluate TRACI using the open-weight LLM DeepSeek-V3 (Liu et al., 2024a) and three cost-effective ChatGPT variants: GPT-3.5 (Brown et al., 2020), GPT-4o-mini (Achiam et al., 2023) and GPT-4.1-mini (OpenAI, 2025). The corresponding results are reported in Figure 4. On the Cora dataset, the variant of TRACI equipped with GPT-4o-mini generally yields the best overall performance, while the GPT-3.5 based model also delivers impressive and competitive results. Although DeepSeek-V3 shows comparatively lower performance overall, it remains competitive on certain metric(e.g., NMI of $0.4295 \pm 0.0237$ on CiteSeer).

## 4.5 CASE STUDY

To facilitate a more intuitive understanding of TRACI, we present two illustrative case studies. The first case, showing in Figure 4 highlights the contribution of LLM-based expansion. Specifically, we examine node 25, a sample from the minority class *Reinforcement Learning* in the Cora dataset under an imbalance ratio of 10. Without LLM expansion, this node is misclassified into the majority class *Genetic Algorithms*. In contrast, our approach leverage the LLM's strong textual understanding to generate augmented textual views for each sample. As a result, the LLM-guided view enables the correct identification of node 25 as belonging to the intended minority class. This positive feedback from the LLM further boosts the learning of minority-aware representations under class-imbalanced settings. The second case study, involving CiteSeer's minority-class node 141, labeled as *Human Computer Interaction*, demonstrates text-guided long-tailed mixup. Our method assigns it higher mixing weight based on LLM-assigned semantic relevance, enabling correct clustering, whereas equal weighting results in misclassification into the majority class *Information Retrieval*. In conclusion, these two representative cases strongly demonstrate the reliability and interpretability of TRACI in learning minority-aware representations under long-tailed class distributions.

## 5 CONCLUSION

In this work, we propose a novel text-guided group mixup framework with canonical correlation alignment to address the challenge of imbalanced text-attributed graph clustering. TRACI leverages large language models (LLMs) to generate semantically enriched text augmentations, enhancing representation consistency across views. A text-guided mixup strategy is employed to adaptively prioritize minority samples based on LLM-derived semantic relevance. Furthermore, LLM-generated ranking signals are utilized to refine the representations of boundary nodes. Extensive experiments demonstrate the effectiveness of TRACI under long-tailed imbalanced conditions. In future work, we plan to extend our method to cluster multi-modal graphs that integrate diverse information sources (e.g., text and images) (Fang et al., 2025), and further explore its scalability and applicability to real-world, large-scale imbalanced datasets.

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

# A DATASETS

To evaluate TRACI under class-imbalanced scenarios, we construct several novel graphs with varying imbalance ratios for these four acknowledged datasets, Cora (McCallum et al., 2000), Cite-Seer (Giles et al., 1998), WikiCS (Mernyei & Cangea, 2020) and PubMed (Sen et al., 2008). The sampling criterion aims to ensure that nodes in each class follow a long-tailed distribution while preserving the overall connectivity as much as possible. Specifically, nodes with higher degrees are retained, whereas those with lower degrees are removed accordingly. Detailed statistics are provided in Table 5.

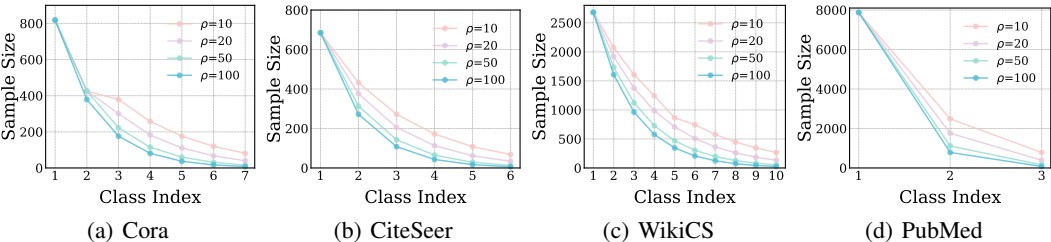

| (a) Cora | (b) CiteSeer | (c) WikiCS | (d) PubMed |

Figure 3: Sample size of per class in the Cora, CiteSeer, WikiCS and Pubmed with varying imbalance ratios ($\rho$=10, 20, 50 and 100). It is evident that the sample size follows a long-tailed distribution.

Table 5: Statistics of datasets with varying imbalance ratios.

| Imbalance | $\rho = 10$ | | $\rho = 20$ | | $\rho = 50$ | | $\rho = 100$ | | | |
| Dataset | #Nodes | #Edges | #Nodes | #Edges | #Nodes | #Edges | #Nodes | #Edges | #Features | #Clusters |
|---|---|---|---|---|---|---|---|---|---|---|
| Cora | 2,258 | 4,602 | 1,945 | 3,878 | 1,688 | 3,178 | 1,516 | 2,801 | 384 | 7 |
| CiteSeer | 1,737 | 2,791 | 1,476 | 2,375 | 1,248 | 2,023 | 1,131 | 1,807 | 384 | 6 |
| WikiCS | 10,848 | 214,593 | 9,117 | 206,770 | 7,496 | 186,880 | 6,645 | 172,124 | 384 | 10 |
| PubMed | 11,152 | 31,890 | 10,028 | 27,704 | 9,145 | 23,495 | 8,740 | 21,381 | 384 | 3 |

# B BASELINES

The compared approaches are comprehensively described as follows:

- DMoN (Tsitsulin et al., 2023) is an unsupervised clustering framework for attributed graphs based on GNNs. It enables end-to-end differentiable optimization of cluster assignments through modularity maximization combined with collapse regularization.

- Dink-Net (Liu et al., 2023a) proposes a self-supervised clustering approach designed to scale to large graphs. Specifically, it jointly models representation learning and clustering by pushing apart different clusters and pulling nodes closer to their assigned clusters, using an augmentation-based discriminative strategy.

- HSAN (Liu et al., 2023b) is a contrastive deep graph clustering framework tailored to handle both hard positive and hard negative sample pairs. It introduces a similarity measure that jointly considers both attribute and structural information.

- S$^3$GC (Devvrit et al., 2022) leverages contrastive learning in combination with Graph Neural Networks and node features, making it well-suited for large-scale datasets.

- DGCluster (Bhowmick et al., 2024) proposes a novel framework that uses pairwise soft memberships between nodes to address the graph clustering problem through modularity maximization. Its computational complexity scales linearly with the size of the graph, making it well-suited for large-scale datasets.

- MAGI (Liu et al., 2024b) is a contrastive learning method based on modularity maximization. It forms positive pairs from nodes within the same module and negative pairs from nodes belonging to different modules, thereby effectively leveraging the graph structure.

- IsoSEL (Sun et al., 2025) proposes a Lorentz tree contrastive learning framework with isometric augmentation to refine the deep partitioning tree in hyperbolic space, while also incorporating attribute information.

For the above baselines, we retrain each model on our constructed datasets and report their performance averaged over five runs to ensure a fair comparison.

## C  EVALUATION METRICS

In this work, we use three widely used clustering metrics: accuracy (ACC), normalized mutual information (NMI) and macro F1 score as metrics to evaluate comprehensively clustering performance of methods.

- **ACC** is a commonly used metric for evaluating classification performance. In the context of unsupervised clustering, the predicted clusters must first be aligned with the ground truth labels using the Hungarian algorithm, based on the confusion matrix $\mathbf{C} \in \mathbb{R}^{K \times K}$, where $K$ is the number of classes and $C_{i,j}$ denotes the number of samples with ground truth label $i$ and predicted label $j$. Specifically, ACC is defined as:

$$\text{ACC} = \frac{\sum_{i=1}^{K} C_{i,i}}{\sum_{i=1}^{K} \sum_{j=1}^{K} C_{i,j}}. \tag{14}$$

- **NMI** calculates consistency between the predicted and true labels. Specifically, given two clustering results $X = (X_1, X_2, ..., X_r)$ and $Y = (Y_1, Y_2, ..., Y_s)$,

$$\text{NMI} = \frac{I(X,Y)}{\max\{H(X), H(Y)\}}, \tag{15}$$

where $I(X,Y)$ is the mutual information between X and Y, $H(X)$ and $H(Y)$ are the entropy of X and Y respectively.

- **F1 score** is a widely used metric for evaluating multi-class classification performance. We compute the macro F1 score by taking the arithmetic mean of the per-class F1 scores, treating all classes equally regardless of their support. For a dataset with $K$ classes, the macro F1 score is calculated as:

$$\text{F1 score} = \frac{\sum_{k=1}^{K} \text{F1 score}_k}{K}, \tag{16}$$

where the F1 score for each class $k$ is given by:

$$\text{F1 score}_k = \frac{2\text{TP}}{2\text{TP} + \text{FP} + \text{FN}}, \tag{17}$$

where TP denotes the number of samples correctly predicted as positive; FP represents the number samples incorrectly predicted as positive; FN is the number of samples incorrectly predicted as negative; and TN refers to the number of samples correctly predicted as negative.

## D  PROOF OF THEOREM 3.1

In this section, we present a detailed proof of Theorem 3.1, which is restated below for completeness.

***Proof of Theorem 3.1***: Without loss of generality, we assume the encoder $f_\Theta$ is $L$-Lipschitz continuous. Let $\mathcal{F}$ be the hypothesis class of $L$-Lipschitz encoders. For contrastive loss $\mathcal{L}_{cl}$ and $\mathcal{L}_{mixup}$, we can decompose them in the following formula:

$$\mathcal{L}_{cl} = -\frac{1}{N}\sum_{i=1}^{N} \log \frac{e^{\theta(\mathbf{z}_i^{(1)}, \mathbf{z}_i^{(2)})/\tau}}{e^{\theta(\mathbf{z}_i^{(1)}, \mathbf{z}_i^{(2)})/\tau} + \sum_{\mathbf{z}_i^-} e^{\theta(\mathbf{z}_i^{(1)}, \mathbf{z}_i^-)/\tau}} = \frac{1}{N}\sum_{i=1}^{N} \ell_{cl}\big(f_\Theta(\mathbf{x}_i)\big), \tag{18}$$

and

$$\mathcal{L}_{\text{mixup}} = -\frac{1}{M}\sum_{m=1}^{M} \log \frac{e^{\theta(\mathbf{h}_m^{(1)}, \mathbf{h}_m^{(2)})/\tau}}{e^{\theta(\mathbf{h}_m^{(1)}, \mathbf{h}_m^{(2)})/\tau} + \sum_{\mathbf{h}_m^-} e^{\theta(\mathbf{h}_m^{(1)}, \mathbf{h}_m^-)/\tau}} = \frac{1}{M}\sum_{m=1}^{M} \ell_{\text{mixup}}\big(f_\Theta(\mathbb{G}_m)\big), \tag{19}$$

where $\theta(\mathbf{z}_i^{(1)}, \mathbf{z}_i^{(2)}) = \frac{(\mathbf{z}_i^{(1)})^T \cdot \mathbf{z}_i^{(2)}}{||\mathbf{z}_i^{(1)}|| \cdot ||\mathbf{z}_i^{(2)}||}$ calculates the cosine similarity between two vectors.

*For* $\mathcal{L}_{CL}$, the empirical Rademacher complexity is:

$$\mathcal{R}(\mathcal{L}_{\text{cl}}) = \mathbb{E}_\sigma \left[ \sup_{f \in \mathcal{F}} \frac{1}{N} \sum_{i=1}^{N} \sigma_i \ell_{\text{cl}}\big(f(\mathbf{x}_i)\big) \right], \tag{20}$$

where $\ell_{\text{cl}}(f_\Theta(\mathbf{x}_i))$ is the contrastive loss for the positive pair $(\mathbf{z}_i^{(1)}, \mathbf{z}_i^{(2)})$. Subsequently, we try to bound the empirical Rademacher complexity for $\ell_{\text{cl}}$. We define a function $\phi$ as the normalization-score operation for positive-negative sample pairs:

$$\phi\left(\theta(\mathbf{z}_i, \mathbf{z}_i^+), \theta(\mathbf{z}_i, \mathbf{z}^-)\right) = -\log \frac{e^{\theta(\mathbf{z}_i, \mathbf{z}_i^+)/\tau}}{e^{\theta(\mathbf{z}_i, \mathbf{z}_i^+)/\tau} + \sum_{\mathbf{z_i}^-} e^{\theta(\mathbf{z}_i, \mathbf{z}_i^-)/\tau}}. \tag{21}$$

We then prove $\phi$ is $\frac{\sqrt{2}}{\tau}$-Lipschitz continuous.

Define vector $\mathbf{v} = \left(\theta(\mathbf{z}_i, \mathbf{z}_i^+), \theta(\mathbf{z}_i, \mathbf{z}_j)_{j \neq i}, \theta(\mathbf{z}_i, \mathbf{z}_j^-)_{j \neq i}\right) \in \mathbb{R}^{2N-1}$, then $\phi$ can be written as a function of $\mathbf{v}$

$$\phi(v) = -\log \frac{e^{v_0/\tau}}{\sum_{j=0}^{2N-1} e^{v_j/\tau}} = \log \sum_{j=0}^{2N-1} e^{(v_j - v_0)/\tau}. \tag{22}$$

The derivative of $\phi$ with respect to $v_0$ is

$$\nabla_{v_0} \phi = \frac{1}{\tau} \cdot \left( \frac{e^{v_0/\tau}}{\sum_{j=0}^{2N-1} e^{v_j/\tau}} - 1 \right) \triangleq \frac{1}{\tau} \cdot p_0, \tag{23}$$

and the derivative of $\phi$ with respect to $v_j (j > 0)$ is

$$\nabla_{v_j} \phi = \frac{1}{\tau} \cdot \frac{e^{v_j/\tau}}{\sum_{j=0}^{2N-1} e^{v_j/\tau}} \triangleq \frac{1}{\tau} \cdot p_j. \tag{24}$$

Thus, the $\ell_2$-norm of $\phi$ satisfies: $||\nabla_{\mathbf{v}} \phi||_2 = \frac{1}{\tau} \sqrt{(1 - p_0)^2 + \sum_{j \geq 1} p_j^2} \leq \frac{\sqrt{2}}{\tau}$, where the inequality holds because $\sum_{j \geq 0} p_j = 1$, which implies $\sum_{j \geq 0} p_j^2 \leq 1$. Therefore, $\phi$ is proved to be $\frac{\sqrt{2}}{\tau}$-Lipschitz continuous. Subsequently, by applying the contraction lemma with $f \in \mathcal{F}$, we can bound the empirical complexity for $\mathcal{L}_{\text{cl}}$ as follows:

$$\mathcal{R}(\mathcal{L}_{\text{CL}}) = \mathbb{E}_\sigma \left[ \sup_{f \in \mathcal{F}} \frac{1}{N} \sum_{i=1}^{N} \sigma_i \ell_{\text{cl}}\big(f(\mathbf{x}_i)\big) \right]$$

$$\leq \frac{\sqrt{2}L}{\tau} \cdot \mathbb{E}_\sigma \left[ \sup_{f \in \mathcal{F}} \frac{1}{N} \sum_{i=1}^{N} \sigma_i ||\mathbf{z}_i|| \right]$$

$$\leq \frac{\sqrt{2}L}{\tau} \mathbb{E}_\sigma \left[ \sup_{f \in \mathcal{F}} \frac{1}{N} \sum_{i=1}^{N} \sigma_i \right]$$

$$\leq \frac{2L}{\tau} \sqrt{\frac{\log N}{N}}, \tag{25}$$

where the first equality holds since $\phi$ is $\frac{\sqrt{2}}{\tau}$-Lipschitz continuous and $f \in \mathcal{F}$ is $L$-Lipschitz continuous, the second inequality follows from the normalization condition $||\mathbf{z}_i|| \leq 1$ is normalized, and the third inequality holds due to Massart's theorem, which states that $\mathbb{E}_\sigma \left[ \sup_{f \in \mathcal{F}} \frac{1}{N} \sum_{i=1}^{N} \sigma_i \right] \leq \sqrt{\frac{2 \log N}{N}}$.

*For* $\mathcal{L}_{mixup}$, group embeddings are derived from $\mathbf{h}_m = \sum_{n \in \mathbb{G}_m} s_{m,n} \mathbf{z}_n$ reduces the effective sample size from $N$ to $M$, then the empirical Rademacher complexity is

$$\mathcal{R}(\mathcal{L}_{\text{mixup}}) = \mathbb{E}_\sigma \left[ \sup_{f \in \mathcal{F}} \frac{1}{M} \sum_{m=1}^{M} \sigma_m \ell_{\text{mixup}}\big(f(\mathbb{G}_m)\big) \right]. \tag{26}$$

where $\sigma_i \in \{\pm 1\}$ are Rademacher variables, and $\ell_{\text{mixup}}$ is the contrastive loss for the pair $(h_m^{(1)}, h_m^{(2)})$. Subsequently, we prove the empirical Rademacher complexity is bounded for $\mathcal{L}_{mixup}$. Specifically, since $\sum_{n \in \mathbb{G}_m} s_{m,n} = 1$, then we have

$$\|\mathbf{h}_m\| \leq \sum_{n \in G_m} s_{mn} \|\mathbf{z}_n\| \leq \|\mathbf{s}_m\|_1 \cdot \|\mathbf{z}\|_\infty \leq 1, \tag{27}$$

where the first inequality follows from the triangle inequality for norms, the second inequality holds by Hölder inequality, and the third inequality holds since $\|\mathbf{z}_n\| \leq 1$. Similar to Equation 25, we can derive the bound for $\mathcal{L}_{\text{mixup}}$:

$$\mathcal{R}(\mathcal{L}_{\text{mixup}}) = \mathbb{E}_\sigma \left[ \sup_{f \in \mathcal{F}} \frac{1}{M} \sum_{m=1}^M \sigma_m \ell_{\text{mixup}}(f(\mathbf{h}_m)) \right]$$

$$\leq \frac{\sqrt{2}L}{\tau} \cdot \mathbb{E}_\sigma \left[ \sup_{f \in \mathcal{F}} \frac{1}{M} \sum_{m=1}^M \sigma_m \|\mathbf{h}_m\|_2 \right] \tag{28}$$

$$\leq \frac{\sqrt{2}L}{\tau} \cdot \mathbb{E}_\sigma \left[ \sup_{f \in \mathcal{F}} \frac{1}{M} \sum_{m=1}^M \sigma_m \right] \tag{29}$$

$$\leq \frac{2L}{\tau} \sqrt{\frac{\log M}{M}}. \tag{30}$$

$\sqrt{\frac{\log M}{M}} \ll \sqrt{\frac{\log N}{N}}$ since $M \ll N$ under imbalanced scenarios, then we can derive that the empirical complexity for $\mathcal{L}_{\text{mixup}}$ and $\mathcal{L}_{\text{cl}}$ satisfies that $\mathcal{R}(\mathcal{L}_{\text{mixup}}) \leq \mathcal{R}(\mathcal{L}_{\text{cl}})$, implying a tighter generalization error bound for $\mathcal{L}_{\text{mixup}}$ compared to $\mathcal{L}_{\text{cl}}$.

According to Theorem 26.5 in (Shalev-Shwartz & Ben-David, 2014), we can derive the generalization bound under the condition that $\ell_{\text{cl}}$ and $\ell_{\text{mixup}}$ are bounded, which is ensured by the normalization of $\mathbf{z}$ in the embedding space. Then, with probability at least $1 - \delta$, for any encoder $f \in \mathcal{F}$, there exists a constant $C$ such that the generalization error bound satisfies:

$$\mathcal{E} - \mathcal{E}^* \leq 2\mathcal{R} + C\sqrt{\frac{\log(1/\delta)}{N}}, \tag{31}$$

where $\mathcal{R}$ is the empirical Rademacher complexity for loss function $\mathcal{L}$. By substituting the loss function with $\mathcal{L}$cl and $\mathcal{L}$mixup, we can derive the respective generalization error bounds for the above contrastive losses, which are expressed as follows:

$$\mathcal{E}(\mathcal{L}_{\text{mixup}}) - \mathcal{E}^*(\mathcal{L}_{\text{mixup}}) \leq C(\sqrt{\frac{\log M}{M}} + \sqrt{\frac{\log(1/\delta)}{N}}), \tag{32}$$

$$\mathcal{E}(\mathcal{L}_{\text{cl}}) - \mathcal{E}^*(\mathcal{L}_{\text{cl}}) \leq C(\sqrt{\frac{\log N}{N}} + \sqrt{\frac{\log(1/\delta)}{N}}), \tag{33}$$

where $C$ is a constant. Thus, we conclude that $\mathcal{L}_{\text{mixup}}$ yields a significantly tighter bound on the generalization error in comparison to $\mathcal{L}_{\text{cl}}$. $\square$

## E  PROMPT DESIGN

Table 6 presents the prompts used to generate textual augmented views via the LLM.

Table 7 presents the prompts provided to the LLM for different purposes, including *Group Mixup*, *Concept Induction*, and *Ranking Guidance*.

## F  IMPLEMENTATION DETAILS

The implementation details are organized into two parts: the first describes the process of updating cluster centroids and assigning soft labels using the smooth k-means approach proposed by (He, 2024b), while the second outlines the hyperparameter settings of TRACI for different datasets.

Table 6: Prompts for *Data Expansion*.

| Augmentation | Prompt |
|---|---|
| **Technical** | You are an AI assistant specializing in text optimization. Please rewrite the following article while preserving its core ideas. *Requirements:* 1. Use formal academic language with domain-specific terminology. 2. Maintain strict factual consistency with the original content. |
| **Colloquial** | You are an AI assistant specializing in text optimizing. Please simplify this article for non-experts while retaining key information. *Requirements:* 1. Avoid technical jargon. 2. Use short sentences and everyday vocabulary. |

Table 7: Prompts for *Group Mixup*, *Concept Induction* and *Ranking Guidance*.

| Usage of the LLM | Prompt |
|---|---|
| **Group Mixup** | You are an AI assistant specializing in semantic analysis. Please evaluate the contribution and confidence scores of each text with respect to the whole cluster. *Requirements for contribution scores:* 1. The contribution score should range from 0.00 to 1.00. A lowest score of 0.00 indicates the lowest contribution while 1.00 reflects the highest contribution. 2. Consider its semantic relevance to the cluster, the density it contains, its conceptual representativeness, and its contextual coherence with other texts. 3. Derive an overall contribution score by synthesizing these individual evaluations. *Requirements for confidence scores:* 1. Fall within the range of 0.00 to 1.00. 2. Evaluate the accuracy and credibility of the contribution score. |
| **Concept Induction** | You are an AI assistant specializing in topic modeling. Please examine the core themes and contextual elements within the input texts to generate a concise, accurate topic name. *Requirements:* 1. Analyze the commonalities and core content of these samples. 2. Provide a concise summary of the cluster's theme. 3. Output the theme as a short name. |
| **Ranking Guidance** | You are an AI assistant specializing in text prediction. Please analyze the content to determine the most relevant topic cluster it belongs to. *Requirements:* 1. Consider comprehensively which cluster this article most likely belongs to. 2. The optimal clusters are ¡Topics induced by Concept Induction¿. 3. Answer the cluster number directly. |

**Smooth K-means.** Inspired by the problem of imbalanced clustering, (He, 2024b) proposes a novel method based on k-means with a Boltzmann operator, which we adopt in place of the traditional hard k-means for clustering. Specifically, the cluster centroids $\mathbf{c}_k$ and the weighted cluster assignments $\omega n, k$ are defined as follows:

$$\omega_{n,k} = \frac{e^{-\gamma \cdot d_{n,k}}}{\sum_{k=1}^{K} e^{-\gamma \cdot d_{n,k}}} \left[ 1 - \gamma \left( d_{n,k} - \frac{\sum_{k=1}^{K} d_{n,k} e^{-\gamma \cdot d_{n,k}}}{\sum_{k=1}^{K} e^{-\gamma \cdot d_{n,k}}} \right) \right], \quad (34)$$

and

$$\mathbf{c}_k = \frac{\sum_n w_{n,k} \mathbf{z}_n}{\sum_n w_{n,k}}, \tag{35}$$

where $\omega_{n,k}$ denotes the weight of the $n$-th sample with respect to the $k$-th cluster such that $\sum_k \omega_{n,k} = 1$, $d_{n,k}$ is the distance between the $n$-th sample and the $k$-th cluster, $\gamma$ is a smoothing hyperparameter, and $\mathbf{z}_n$ represents the embedding of the $n$-th sample.

**Hyperparamter Settings.** Additional hyperparameter details are thoroughly described in Table 8.

Table 8: Hyperparameter settings. *Hidden Dimensions* refers to the dimension of the GCN encoder in the latent space. $\text{lr}_1$ and $\text{lr}_2$ denote the learning rates used during the warmup and fine-tuning stages, respectively. $\alpha$ and $\beta$ are hyperparameters that weight different loss components, while $\gamma$ controls the strength of the smooth k-means term. $\tau_1$ and $\tau_2$ are the temperature parameters for the mixup loss and ranking loss, respectively. $M$ represents the number of groups.

| Dataset | Hidden Dimensions | $\text{lr}_1$ | $\text{lr}_2$ | $\alpha$ | $\beta$ | $\gamma$ | $\tau_1$ | $\tau_2$ | $M$ |
|---|---|---|---|---|---|---|---|---|---|
| **Cora** | $[64]$ | 0.0005 | 0.0001 | 0.1 | 0.9 | 10 | 0.5 | 0.01 | 100 |
| **CiteSeer** | $[128, 64]$ | 0.0001 | 0.0001 | 0.2 | 0.9 | 10 | 0.9 | 0.01 | 100 |
| **WikiCS** | $[256, 128]$ | 0.0001 | 0.0001 | 0.1 | 0.9 | 16 | 0.9 | 0.09 | 500 |
| **PubMed** | $[128, 64]$ | 0.0005 | 0.0001 | 0.1 | 0.9 | 10 | 0.5 | 0.01 | 500 |

## G PERFORMANCE ON MORE IMBALANCED SCENARIOS

Previously, we have evaluated the performance of TRACI against baselines under imbalance ratios of $\rho = 10$ and $\rho = 20$. In this section, we explore more extreme scenarios with imbalance ratios of $\rho = 50$ and $\rho = 100$, where the minority class contains substantially fewer samples, making the imbalanced graph clustering considerably more challenging. As shown in Table 9, TRACI consistently demonstrates competitive overall performance across these settings, further highlighting its robustness and effectiveness under severe class imbalance conditions.

Table 9: Clustering performance of TRACI compared to baseline methods under more imbalanced scenarios ($\rho = 50$ and 100).

| Imbalance | Dataset | Metric | DMoN | Dink-Net | HSAN | S³GC | DGCluster | MAGI | IsoSEL | TRACI |
|---|---|---|---|---|---|---|---|---|---|---|
| | Cora | ACC | $53.89_{\pm 6.54}$ | $47.20_{\pm 1.84}$ | $47.39_{\pm 1.61}$ | $47.57_{\pm 1.69}$ | $56.53_{\pm 5.56}$ | $48.85_{\pm 0.70}$ | $\mathbf{62.32}_{\pm 2.44}$ | $\underline{58.73}_{\pm 4.79}$ |
| | | NMI | $\mathbf{48.13}_{\pm 0.48}$ | $46.30_{\pm 0.53}$ | $44.71_{\pm 0.39}$ | $43.37_{\pm 0.43}$ | $44.72_{\pm 1.27}$ | $46.00_{\pm 0.42}$ | $44.96_{\pm 2.32}$ | $\underline{47.44}_{\pm 1.22}$ |
| | | F1 | $39.78_{\pm 3.26}$ | $37.33_{\pm 1.90}$ | $40.24_{\pm 1.17}$ | $37.57_{\pm 1.57}$ | $\underline{42.48}_{\pm 3.22}$ | $39.79_{\pm 0.43}$ | $38.41_{\pm 4.31}$ | $\mathbf{42.62}_{\pm 4.41}$ |
| | CiteSeer | ACC | $54.87_{\pm 5.55}$ | $55.46_{\pm 3.54}$ | $50.64_{\pm 0.96}$ | $47.69_{\pm 3.34}$ | $53.89_{\pm 3.65}$ | $56.36_{\pm 0.44}$ | $\underline{57.02}_{\pm 8.10}$ | $\mathbf{59.63}_{\pm 9.22}$ |
| | | NMI | $36.40_{\pm 1.83}$ | $38.96_{\pm 1.44}$ | $37.99_{\pm 0.95}$ | $37.24_{\pm 1.73}$ | $34.08_{\pm 0.19}$ | $\underline{39.21}_{\pm 0.29}$ | $34.72_{\pm 1.43}$ | $\mathbf{39.97}_{\pm 2.12}$ |
| $\rho = 50$ | | F1 | $39.28_{\pm 5.89}$ | $39.37_{\pm 4.18}$ | $39.93_{\pm 1.21}$ | $38.04_{\pm 2.69}$ | $25.65_{\pm 3.00}$ | $\mathbf{45.32}_{\pm 0.15}$ | $34.09_{\pm 6.67}$ | $\underline{43.93}_{\pm 8.83}$ |
| | WikiCS | ACC | $37.79_{\pm 2.82}$ | $54.97_{\pm 5.64}$ | $54.37_{\pm 6.02}$ | $\underline{56.18}_{\pm 1.20}$ | $47.93_{\pm 5.49}$ | $49.46_{\pm 1.85}$ | | $\mathbf{63.19}_{\pm 5.81}$ |
| | | NMI | $31.20_{\pm 1.71}$ | $46.77_{\pm 0.90}$ | $47.34_{\pm 1.58}$ | $41.61_{\pm 0.43}$ | $47.17_{\pm 1.09}$ | $\underline{47.69}_{\pm 0.16}$ | OOM | $\mathbf{48.98}_{\pm 1.34}$ |
| | | F1 | $28.41_{\pm 4.28}$ | $40.38_{\pm 2.47}$ | $\underline{41.13}_{\pm 2.13}$ | $30.91_{\pm 0.87}$ | $39.96_{\pm 3.80}$ | $39.88_{\pm 0.82}$ | | $\mathbf{49.79}_{\pm 4.54}$ |
| | Cora | ACC | $50.63_{\pm 4.62}$ | $49.09_{\pm 3.14}$ | $45.83_{\pm 3.53}$ | $45.77_{\pm 3.37}$ | $\underline{55.40}_{\pm 5.75}$ | $46.21_{\pm 0.90}$ | $52.61_{\pm 7.74}$ | $\mathbf{62.18}_{\pm 2.40}$ |
| | | NMI | $\mathbf{45.03}_{\pm 2.56}$ | $39.40_{\pm 1.33}$ | $43.45_{\pm 1.69}$ | $42.03_{\pm 1.04}$ | $41.63_{\pm 0.64}$ | $\underline{45.01}_{\pm 0.60}$ | $41.84_{\pm 2.97}$ | $43.64_{\pm 1.63}$ |
| | | F1 | $36.33_{\pm 2.11}$ | $35.89_{\pm 0.84}$ | $36.83_{\pm 0.84}$ | $34.82_{\pm 1.15}$ | $35.65_{\pm 3.30}$ | $\underline{37.02}_{\pm 0.46}$ | $27.68_{\pm 6.41}$ | $\mathbf{37.93}_{\pm 2.07}$ |
| | CiteSeer | ACC | $56.22_{\pm 4.03}$ | $46.08_{\pm 3.06}$ | $45.18_{\pm 2.50}$ | $47.04_{\pm 3.77}$ | $\underline{56.78}_{\pm 3.81}$ | $56.76_{\pm 0.57}$ | $56.62_{\pm 8.49}$ | $\mathbf{57.24}_{\pm 0.93}$ |
| $\rho = 100$ | | NMI | $37.22_{\pm 1.89}$ | $35.69_{\pm 1.21}$ | $34.39_{\pm 1.96}$ | $36.61_{\pm 0.73}$ | $33.27_{\pm 0.81}$ | $\mathbf{39.41}_{\pm 0.45}$ | $32.60_{\pm 0.84}$ | $\underline{39.15}_{\pm 0.33}$ |
| | | F1 | $38.61_{\pm 4.81}$ | $34.65_{\pm 4.71}$ | $35.17_{\pm 1.66}$ | $36.31_{\pm 2.53}$ | $25.53_{\pm 3.78}$ | $\mathbf{43.99}_{\pm 0.23}$ | $31.34_{\pm 6.08}$ | $\underline{41.72}_{\pm 0.27}$ |
| | WikiCS | ACC | $41.35_{\pm 3.66}$ | $44.64_{\pm 2.07}$ | $46.89_{\pm 5.73}$ | $\underline{49.58}_{\pm 1.65}$ | $41.44_{\pm 3.59}$ | $48.98_{\pm 3.03}$ | | $\mathbf{50.78}_{\pm 1.03}$ |
| | | NMI | $33.45_{\pm 1.81}$ | $43.07_{\pm 1.00}$ | $45.36_{\pm 1.87}$ | $36.31_{\pm 0.52}$ | $45.01_{\pm 0.72}$ | $\mathbf{47.41}_{\pm 0.21}$ | OOM | $\underline{46.47}_{\pm 0.87}$ |
| | | F1 | $28.87_{\pm 4.10}$ | $32.71_{\pm 1.45}$ | $34.82_{\pm 2.07}$ | $17.87_{\pm 0.37}$ | $32.34_{\pm 0.69}$ | $\underline{36.23}_{\pm 1.15}$ | | $\mathbf{40.65}_{\pm 1.64}$ |

## H ALGORITHM

We present the overall algorithm of TRACI in Algorithm 1.

## I CASE STUDY

Here, we provide a case to illustrate the effect of TRACI.

---

**Algorithm 1:** The algorithm of TRACI

---

**Input:** $\mathcal{G} = (\mathbf{A}, \mathcal{D})$, GNN encoder $\mathcal{F}_\Theta$, training epoch $T$, the LLM execution interval $T'$,
learning rates $\text{lr}_1$ and $\text{lr}_2$, number of selected nodes $n$, number of groups $M$.

1 Augment $\mathcal{D}$ into $\mathcal{D}^{(1)}$ and $\mathcal{D}^{(2)}$, yielding $\mathcal{G}^{(1)}$ and $\mathcal{G}^{(2)}$, respectively ;

/* Warmup */

2 **for** $t \leftarrow 1$ **to** $T$ **do**

3     Encode: $\mathbf{Z}^{(1)} = \mathcal{F}(\mathcal{G}^{(1)})$ and $\mathbf{Z}^{(2)} = \mathcal{F}(\mathcal{G}^{(2)})$;

4     **if** $t \% T' == 0$ **then**

5        $\mathbf{S}^{(1)}, \mathbf{S}^{(2)} \leftarrow$ GetWeightMatrix $(\mathcal{D}^{(1)}, \mathcal{D}^{(2)})$ ;

6        Obtain the group-level synthetic embeddings $\mathbf{H}^{(i)} = (\mathbf{h}_m^{(1)})_{m=1}^M$ through Eq. 3 for
       $i = 1, 2$;

7     Calculate the warmup loss $\mathcal{L}_{\text{warm}}$ in Eq. 5 using $\mathcal{L}_{\text{corr}}$ in Eq. 1 and $\mathcal{L}_{\text{mixup}}$ in Eq. 4;

8     Update: $\Theta \leftarrow \Theta - \text{lr}_1 \cdot \nabla \mathcal{L}_{\text{warm}}$;

/* Fine-tuning with Ranking Guidance */

9 Obtain ranking guidance in Eq. 8 with challenge nodes $\mathcal{S}$;

10 Update $\mathbf{X}$ via mean pooling over nodes in $\mathcal{C}_{\text{LLM}}$;

11 **for** $t \leftarrow 1$ **to** $T$ **do**

12     Encode: $\mathbf{Z}^{(1)} = \mathcal{F}(\mathcal{G}^{(1)})$ and $\mathbf{Z}^{(2)} = \mathcal{F}(\mathcal{G}^{(2)})$;

13     Obtain group embeddings similar to line 4 to 6;

14     Calculate the fine-tuning loss $\mathcal{L}_{\text{fine}}$ in Eq. 11 using $\mathcal{L}_{\text{corr}}$ in Eq. 1, $\mathcal{L}_{\text{mixup}}$ in Eq. 4, $\mathcal{L}_{\text{rank}}$ in
    Eq. 10;

15     Update: $\Theta \leftarrow \Theta - \text{lr}_2 \cdot \nabla \mathcal{L}_{\text{fine}}$;

**Output:** Final cluster assignments.

16 **Function** GetWeightMatrix $(\mathcal{D}^{(1)}, \mathcal{D}^{(2)})$:

17     Randomly partition the samples into $M$ groups $\mathbb{G} = \{\mathbb{G}_1, \mathbb{G}_2, \cdots, \mathbb{G}_M\}$;

18     **for** $m \leftarrow 1$ **to** $M$ **do**

19        **for** $i \leftarrow 1$ **to** $2$ **do**

20           Query contribution score $b_{mn}^{(i)}$ and confidence score $c_{mn}^{(i)}$ by an LLM through $\mathcal{P}_{\text{mix}}$;

21           Compute correlation-based weight score $s_{mn}^{(i)}$ through Eq. 2;

22     **return** $\mathbf{S}^{(1)}, \mathbf{S}^{(2)}$

Figure 4: Case study of node 25 from the Cora dataset under an imbalance ratio of 10.

## J  COMPUTATIONAL COST

In this work, we propose TRACI, a method that integrates Graph Neural Networks (GNNs) with Large Language Models (LLMs) for graph clustering under class imbalance scenarios. The computational cost associated with the use of LLMs mainly arises from three components: the first is *Data Expansion*, where the LLM is used to generate augmented textual views for nodes; the second, termed *Group Mixup*, leverages the LLM to compute semantic correlation scores between texts within the same group to enhance contextual representations; and the third, referred to as *Ranking Guidance*, involves using the LLM to predict the most likely cluster assignment, thereby providing

feedback to guide the GNN. The total running time of TRACI consists of the time required for LLM inference on queries and the GNN's execution time when incorporating LLM-derived feedback.

To comprehensively evaluate the efficiency of TRACI, we report both its computational cost and runtime, as summarized in Table 10, under imbalance ratios $\rho = 10$ and $\rho = 20$. The computational cost is calculated based on the token-level pricing of GPT-4o-mini for both input and output tokens, while the runtime is estimated using its per-minute rate limit. As expected, larger datasets incur higher cost and longer runtime, which may hinder scalability to extremely large graphs. Notably, the *Expansion* step incurs no additional cost or runtime, as the construction of datasets with varying imbalance ratios allows it to reuse a subset of nodes from the version with a lower $\rho$, thus making it computationally free.

Table 10: Computational cost and Running time of TRACI on datasets under imbalanced settings ($\rho = 10$ and 20).

| Imbalance | Datasets | Computational Cost ($) | | | | Running Time (min) | | | | |
|---|---|---|---|---|---|---|---|---|---|---|
| | | Data Expansion | Group Mixup | Ranking Guidance | Total | Data Expansion | Group Mixup | Ranking Guidance | GNN | Total |
| $\rho = 10$ | Cora | 0.88 | 1.30 | 0.13 | 2.32 | 5.98 | 5.10 | 2.72 | 1.20 | 15.01 |
| | CiteSeer | 0.71 | 1.05 | 0.08 | 1.84 | 4.66 | 4.16 | 1.73 | 1.50 | 12.05 |
| | WikiCS | 7.26 | 8.74 | 0.56 | 16.56 | 66.66 | 45.30 | 13.86 | 3.23 | 129.05 |
| | PubMed | 6.55 | 8.12 | 0.13 | 14.81 | 48.27 | 38.15 | 2.73 | 1.72 | 90.86 |
| $\rho = 20$ | Cora | 0.00 | 1.15 | 0.09 | 1.23 | 0.00 | 4.45 | 1.82 | 1.12 | 7.38 |
| | CiteSeer | 0.00 | 0.90 | 0.04 | 0.95 | 0.00 | 3.59 | 0.98 | 1.50 | 6.07 |
| | WikiCS | 0.00 | 7.47 | 0.37 | 7.84 | 0.00 | 38.61 | 9.43 | 3.97 | 52.01 |
| | PubMed | 0.00 | 7.35 | 0.12 | 7.48 | 0.00 | 34.37 | 2.63 | 1.67 | 38.67 |

# K  LLM USAGE CLARIFICATION

In this study, we use a large language model (LLM) solely to detect grammatical errors and improve sentence clarity. No content is generated automatically beyond these language edits, and all suggested modifications are carefully reviewed by the authors. All scientific aspects, including research hypotheses, experimental design, data analysis, and conclusions, are fully conceived and verified by the authors.

