# OpenReview forum: "Text-guided Group Mixup with Canonical Mining for Imbalanced Graph Clustering"
_ICLR.cc/2026/Conference — ICLR 2026 Conference Withdrawn Submission_

### Official Review · Reviewer_eQ75 · 2025-10-20

**Soundness:** 2
**Presentation:** 3
**Contribution:** 2
**Rating:** 2
**Confidence:** 4

**Summary:**

This paper addresses imbalanced text-attributed graph clustering and proposes TRACI, an LLM-guided framework combining dual-view text augmentation, canonical correlation alignment, and LLM-based group mixup with ranking refinement. Experiments on small academic graphs (Cora, CiteSeer, PubMed, WikiCS) show improved clustering performance under long-tailed settings. However, the method involves heavy LLM usage, lacks runtime and scalability evaluation, and is limited to academic-style text domains.

**Strengths:**

- Clearly identifies and formalizes the imbalanced TAG clustering setting
- The paper is well-written

**Weaknesses:**

- The method involves multiple LLM calls per node (paraphrasing, scoring, ranking), making it computationally and financially impractical for large-scale use.
- All results are on tiny academic datasets (Cora, CiteSeer, PubMed, WikiCS), with no evidence the method works on larger or real-world graphs.
- The paper only estimates cost by token count; there are no measured runtimes or scalability experiments, where I doubt about its scalability.
– The approach assumes texts can be rewritten in “academic” vs. “layman” style, limiting applicability beyond scholarly corpora with narrow domain scope.

**Questions:**

See weaknesses above.

---

### Official Review · Reviewer_XojZ · 2025-10-31

**Soundness:** 2
**Presentation:** 3
**Contribution:** 2
**Rating:** 2
**Confidence:** 4

**Summary:**

This paper notes GNNs’ limitation in text-attributed graph clustering—their assumption of uniform class distribution fails in real imbalanced scenarios. It thus focuses on imbalanced text-attributed graph clustering and proposes the TRACI framework. TRACI’s core relies on LLMs: generating diverse sample views to assign samples into balanced mixed-semantic groups, using LLMs to compute sample correlation scores for minority-aware representations, and enforcing canonical correlations for semantic alignment. Experiments on benchmarks confirm TRACI outperforms state-of-the-art baselines.

**Strengths:**

The results of the experiment are promising.

**Weaknesses:**

1. The paper lacks detailed verification of semantic consistency between LLM-generated augmentations and original texts, which weakens the reliability of the method. While the authors claim that LLM-generated views retain core semantics, they do not specify quantitative metrics (e.g., BERTScore, BLEU, or cosine similarity between original and augmented text embeddings) to measure this consistency.
2. The paper only reports the mean and standard deviation of 5 runs, but does not conduct t-tests or ANOVA to verify whether the performance gap between TRACI and baselines is statistically significant.
3. The experiments only report overall metrics (ACC, F1) but not minority class-specific performance. In imbalanced clustering, some methods may achieve a high overall ACC by sacrificing minority classes. The paper should calculate per-class F1 scores, especially for tail classes, to verify whether TRACI truly improves minority class clustering accuracy.
4. The proof assumes that the GCN encoder f_Θ is L-Lipschitz continuous, but does not explain how to verify this assumption for the used GCN. The paper should supplement the calculation of the encoder’s Lipschitz constant L to confirm that the assumption holds. Without this verification, the theoretical bound becomes a "conditional conclusion" and lacks practical support.

**Questions:**

See Weaknesses.

---

### Official Review · Reviewer_Xbq4 · 2025-10-31

**Soundness:** 3
**Presentation:** 3
**Contribution:** 3
**Rating:** 6
**Confidence:** 4

**Summary:**

Overall, this paper makes a contribution by successfully integrating LLMs into the unsupervised graph clustering pipeline to tackle class imbalance. This paper tackles unsupervised clustering for TAGs in imbalanced scenarios. It uses LLMs to keep minority class information from getting washed out, presenting ways to augment text without losing meaning, recognize and boost minority signals, and get LLM guidance for tricky boundary cases the GNN can't handle.

**Strengths:**

- Using LLMs for the augmented views is a good idea. It effectively dodges the risk of messing up the sentence's meaning, which is a real problem with old methods like word shuffling.
- It's a strong example of how to bring LLMs into unsupervised graph clustering. The fact that it only works for TAGs is a bit of a limitation, but that seems reasonable since the whole method relies on LLM text features.
- The performance jump over other models is large. The ablation study is also well-structured.
- The writing is solid, and the experimental design is well-suited for validating imbalanced clustering.

**Weaknesses:**

- I'm not totally clear on why this has to be a Graph domain problem. It feels like this method could be used directly on any text clustering dataset. I couldn't pinpoint what parts were truly graph-specific.
- Figure 1 is a bit confusing because there's so much going on. The legend items don't seem to perfectly match the elements in the diagram, which made it hard to follow.
- It feels like Figure 4 should have been in the main body of the paper, not the appendix.

**Questions:**

See Weaknesses

---

### Official Review · Reviewer_dwU4 · 2025-11-01

**Soundness:** 3
**Presentation:** 2
**Contribution:** 3
**Rating:** 6
**Confidence:** 4

**Summary:**

This paper addresses class imbalance in real-world graph data through a text-guided group mixup and canonical mining approach. By innovatively leveraging large language models' deep semantic understanding via semantic augmentation and correlation-weighted mechanisms, it effectively enhances minority class recognition while avoiding hallucination risks from direct sample generation. Comprehensive evaluations on multiple benchmark datasets demonstrate its superior clustering accuracy and robustness over existing baselines. This work establishes a new paradigm of using LLMs as semantic guides, providing an effective solution for imbalanced graph data in practical scenarios.

**Strengths:**

1. Moving beyond traditional balanced clustering frameworks, this work focuses on the cutting-edge challenge of "imbalanced text-attributed graph clustering," effectively integrating graph structure learning, textual semantic understanding, and imbalanced learning, demonstrating significant theoretical value and application potential.
2.  Pioneering a novel paradigm that positions large language models as "semantic guides," the proposed approach leverages text-guided group mixup and ranking-guided fine-tuning mechanisms to fully utilize deep semantic understanding while effectively mitigating hallucination risks through consistency filtering.
3. Providing a rigorous mathematical foundation, the work theoretically demonstrates that the proposed group mixup loss achieves a tighter generalization error bound compared to conventional contrastive losses, significantly enhancing the depth and rigor of the research.
4.  Both the model diagrams and the experimental charts exhibit a distinctive style and are visually appealing.

**Weaknesses:**

1. Some descriptions lack sufficient detail, and certain concepts and notations require further explanation, making the reading experience rather challenging.
2. Sensitivity analysis was not performed on the explicitly mentioned hyperparameters in the paper, and the experimental section lacks an overarching framework summarizing the research questions these experiments aim to address.
3. The baseline methods are not introduced in the Related Work section, and there is insufficient analysis and comparison of these methods.

**Questions:**

1. The paper lacks detailed descriptions of the methodological specifics and the conceptual notation employed. The specific issues are as follows:
(1) In Section 3.2, the authors mention that the original test is input into the LLM to generate two different versions of representations. Please specify how these two versions of representations are generated and what the differences are, so as to further argue that this strategy is more semantically-preserving yet stylistically diverse. It is suggested to add further descriptions of the embeddings generated through augmentations in the two augmented views of the original TAG.
(2) In Section 3.3, what is the purpose of calculating centered node embeddings after obtaining node embeddings, and what is the meaning of node embeddings? Please provide an explanation of the symbol Trace in Equation (1). Is it calculating the trace of a matrix?
(3) In Section 3.4, what is the prompt P_mix and how is it designed?
(4) In Section 3.5, there is a lack of specific definition and description of the concept of Boundary Nodes for Querying LLMs. What is P_indu in Concept Induction for Each Cluster and how is it designed? There is also a lack of specific description of the filtering process in line 252.

2. In the Baselines, Dink-Net, HSAN, SGC, DGCLUSTER, MAGI, IsoSEL, and BAT were not mentioned in the related work. It is suggested to analyze and compare them in the related work.

3. It is suggested to add analysis of the temperature hyper-parameter in Equation (10) and the parameters alpha and beta in Equation (11).

4. In terms of writing standards, the following details need to be modified:
(1) Line 052: term frequency-class correlation and topic distribution heterogeneity lack references.
(2) Lines 173 and 174: The statement “As discussed in Section 3.2, the two augmented views are processed through a shared GNN encoder” could not be found in Section 3.2. Should it be Section 3.1?

---

### Note · Authors · 2025-11-29

I have read and agree with the venue's withdrawal policy on behalf of myself and my co-authors.